# 5-Arylidenerhodanines as P-gp Modulators: An Interesting Effect of the Carboxyl Group on ABCB1 Function in Multidrug-Resistant Cancer Cells

**DOI:** 10.3390/ijms231810812

**Published:** 2022-09-16

**Authors:** Ewa Żesławska, Waldemar Tejchman, Annamária Kincses, Gabriella Spengler, Wojciech Nitek, Grzegorz Żuchowski, Ewa Szymańska

**Affiliations:** 1Institute of Biology, Pedagogical University of Krakow, Podchorążych 2, 30-084 Kraków, Poland; 2Department of Medical Microbiology, Albert Szent-Györgyi Medical School, University of Szeged, H-6720 Szeged, Hungary; 3Faculty of Chemistry, Jagiellonian University, Gronostajowa 2, 30-387 Kraków, Poland; 4Chair of Organic Chemistry, Jagiellonian University Medical College, Medyczna 9, 30-688 Kraków, Poland; 5Department of Technology and Biotechnology of Drugs, Jagiellonian University Medical College, Medyczna 9, 30-688 Kraków, Poland

**Keywords:** cancer multidrug resistance, P-glycoprotein, efflux pump inhibition, rhodanine, T-lymphoma cancer cells, crystal structure, molecular docking

## Abstract

Multidrug resistance (MDR) is considered one of the major mechanisms responsible for the failure of numerous anticancer and antiviral chemotherapies. Various strategies to overcome the MDR phenomenon have been developed, and one of the most attractive research directions is focused on the inhibition of MDR transporters, membrane proteins that extrude cytotoxic drugs from living cells. Here, we report the results of our studies on a series newly synthesized of 5-arylidenerhodanines and their ability to inhibit the ABCB1 efflux pump in mouse T-lymphoma cancer cells. In the series, compounds possessing a triphenylamine moiety and the carboxyl group in their structure were of particular interest. These amphiphilic compounds showed over 17-fold stronger efflux pump inhibitory effects than verapamil. The cytotoxic and antiproliferative effects of target rhodanines on T-lymphoma cells were also investigated. A putative binding mode for **11**, one of the most potent P-gp inhibitors tested here, was predicted by molecular docking studies and discussed with regard to the binding mode of verapamil.

## 1. Introduction

Multidrug resistance (MDR) is a severe problem in the treatment of various diseases such as cancer, bacterial, fungal, and parasitic infections. One of the mechanisms of cancer drug resistance is associated with increased drug efflux from cells, mediated by the ATP-binding cassette (ABC) transporters [1]. More than 40 ABC transporters have been identified in humans and classified into seven subfamilies (ABC-A to ABC-G) [2]. Among them, 11 proteins have been reported to be related to MDR, an example of which is P-glycoprotein (P-gp, ABCB1), MDR-associated proteins (MRP, ABCC), and breast cancer resistance protein (BCRP, ABCG2) [3,4]. These proteins and modulation of drug efflux are considered important therapeutic targets in the fight against drug resistance in cancer [5,6,7].

The MDR-ABC transporter most studied in cancer is P-glycoprotein (P-gp, ABCB1). A wide range of compounds with beneficial inhibitory activity toward P-gp have been reported, however, most clinical trials targeting P-gp inhibition have failed, mainly due to the toxicity of the compounds [1,7]. Multiple studies have been performed on P-gp inhibitors, including structure-activity relationship (SAR), crystallization of P-gp in complex with small-molecule compounds, and molecular modeling studies [8,9,10,11]. Most of the collected data suggested that aromatic/hydrophobic interactions could be the key features responsible for binding of the compound to P-gp; however, weak electrostatic interactions (including hydrogen bonding, π-π stacking and cation-π interactions) are also important [8,12].

Rhodanine-derived compounds have been reported to possess a wide spectrum of biological activities, e.g., antidiabetic, antibacterial, antifungal, antimalarial, antitubercular, antiviral and anticancer [13,14,15,16], however, to our knowledge, they have not been studied before as potential modulators of the MDR efflux pump P-gp. On the other hand, closely related in terms of scaffold structure, hydantoin derivatives have been widely reported as inhibitors of P-gp [17,18,19,20,21,22,23]. With this in mind, we have decided to synthesize a series of 5-arylidenerhodanine-based compounds containing additionally one or two aromatic rings (Table 1), and evaluate their inhibition effect of ABCB1 efflux pump. Furthermore, we have tested their cytotoxic and antiproliferative effect on sensitive and ABCB1 overexpressing-resistant mouse T-lymphoma cells. The unusual influence of the carboxyl group present in the structure of the compounds on the bioactivity prompted us to further search for the explanation of the obtained results. Crystal structures of three compounds **3**, **7**, and **11** were determined by the X-ray diffraction method to compare molecular geometries and to search for differences that could be responsible for the observed biological activity/inactivity. Additionally, for the representative of the most active amphiphilic derivative, **11**, the molecular docking studies into the P-gp structure have been performed. The selected physicochemical properties for the whole series of compounds have been estimated as well.

## 2. Results and Discussion

### 2.1. Synthesis

The target compounds (Table 1) have been synthesized according to the procedure described previously for compounds **10–12** [24]. The route of the Knoevenagel condensation of rhodanine or rhodanine-3-carboxyalkyl acids with benzaldehyde derivatives is presented in Figure 1. The rhodanine used for the syntheses was obtained according to the procedure developed by Nencki [25], while the rhodanine-3-carboxyalkyl acids were obtained according to the procedure proposed by Körner [26].

### 2.2. Biological Screening

#### 2.2.1. The Rhodamine 123 Accumulation Assay

All investigated compounds evaluated for their efflux pump modulating effects in the sensitive parental (PAR) and resistant (MDR) mouse T-lymphoma cells overexpressing ABCB1 using the standard rhodamine 123 functional assay at 2 and 20 μM concentrations. The fluorescence activity ratio (FAR) was calculated based on the obtained fluorescence data and shows a measure of the efflux pump modulating properties under the influence of the investigated compounds. Compounds with FAR values below 1 were considered inactive (Table 2).

Among structures containing the 5-(4′-phenylbenzylidene) rhodanine moiety (**1–4**), the modulatory effect was shown only for the derivative **1** at a concentration of 20 μM. An introduction of the carboxyl group to the structure of **1** definitely removed activity (**2–4**). Similar results were observed for compounds **5–8** that possess the 5-(4′-phenylmethoxybenzylidene) rhodanine moiety.

An opposite effect was shown in the case of the 5-(4′-N, N-diphenylaminobenzylidene)rhodanine series **9–12** that proved to be the most potent P-gp modulators among all compounds tested. All four compounds affected the efflux activity of ABCB1 at both concentrations (Table 2). Furthermore, contrary to other series, the introduction of the carboxyl group into the structure of **9** has caused a significant increase in the FAR coefficient and resulted in compounds **10–12** that at 20 μM concentration showed more than 17 times stronger efflux pump inhibitory effects than the reference inhibitor, verapamil (FAR = 4.38). 

#### 2.2.2. Cytotoxicity and Antiproliferative Assays

The presented compounds were also investigated for their cytotoxic and antiproliferative effects in PAR and MDR mouse T-lymphoma cells, using MTT assay to estimate inhibitory concentration of 50% (IC_50_) values (Table 3). Most of the compounds under these conditions did not show significant cytotoxic effects against two tested cell lines, except for **3** (IC_50_ = 13.88 μM and 17.02 μM for PAR and MDR cells, respectively). Antiproliferative activity was observed for all compounds. Interesting results were obtained for **4**, which was not cytotoxic, while its antiproliferative activity was more pronounced in the MDR cell line (IC_50_ = 4.3 μM vs. 8.12 μM), similar to **12** (IC_50_ = 11.16 μM vs. 15.3 μM).

#### 2.2.3. Drug Combination Assay

To evaluate the ability of target rhodanines to resensitize MDR cells to the anticancer drug doxorubicin, interactions between doxorubicin and compounds showing efficient cytotoxic and antiproliferative activities (**1**, **3**, **5**, and **8**) were evaluated using a checkerboard combination assay (Table 4). The level of interactions was expressed by the drug combination index (CI) value, calculated on the basis of IC_50_ data for individual drug or two-drug combinations, and is defined as additivity for the CI value close to 1, synergy for CI < 1, and antagonism CI > 1. Among the compounds tested, **1**, **3**, and **5** displayed an antagonistic effect toward doxorubicin, while interactions between **8** and doxorubicin can be described as nearly additive.

### 2.3. X-ray Studies of Compounds ***3***, ***7***, and ***11***

The molecular geometries in the crystals of the selected compounds (**3**, **7**, and **11**) are presented in Figure 1. Compounds **3** and **7** crystallize with two molecules (labelled A and B) in the asymmetric unit. For all compounds, the isomer *Z* is observed. The 5-bezylidenerhodanine fragment of **7** and **11** is almost planar (Figure 2), while **3** shows a greater deviation from planarity. The biggest differences for the compounds analyzed are observed in the mutual arrangement of aromatic rings. Only for **11** the aromatic rings are arranged almost perpendicular to each other (Appendix A).

The packing of molecules in the crystals is dominated by intermolecular O-H···O hydrogen bonds between two carboxylic groups, which leads to the formation of dimers. The dimers are joined to each other by weak C-H···O and, in the case of **3** and **7**, also by C-H···S interactions. The parameters of these intermolecular interactions are presented in Appendix A. Neither the additional oxygen atom (O4) in **7** nor the additional nitrogen atom (N1) in **11** is involved in any intermolecular hydrogen bonds.

### 2.4. Molecular Modeling

The large and hydrophobic transmembrane interior of P-gp is reported to contain a common binding site for both substrates and small-molecule inhibitors [11]. Competitive inhibitor/substrate binding to the drug-binding cavity is one of the mechanisms suggested for the inhibition of rhodamine 123 efflux, observed in the accumulation assay described above. In an attempt to determine putative binding modes for the described amphiphilic molecules, one of the most active efflux pump modulators, **11**, was docked into the large ABCB1 drug-binding pocket using the induced-fit docking (IFD) protocol implemented in the Schrödinger Suite [27] to mimic binding to a flexible protein. The likely binding interactions between *h*P-gp and **11** were carefully examined and compared with the results of similar docking performed for verapamil, the known first-generation P-gp inhibitor.

Several X-ray structures of murine P-gp crystallized in complex with substrates or inhibitors are known [28,29,30,31], with the protein adopting an inward-facing nucleotide-free conformation that occurs at the initial stage of the ATP-dependent transport cycle [32]. However, so far no such structure of fully human inhibitor-bound ABCB1 is available. On the other hand, a dynamic growth of cryo-EM technology enabled the determination with high-resolution structures of the human ABCB1 transporter in complex with inhibitory antibody fragments and bound to small-molecule substrates or inhibitors [33,34,35]. Most of these structures represent an ‘occluded’ conformation of P-gp, characterized by bent transmembrane helices TM4 and TM10 that form a cytoplasmic gate at the entrance of the drug-binding pocket [33], as well as a reduced distance between two nucleotide-binding domains (NBD), compared to an inward-facing conformation.

Taking into account the above, in our docking studies two following protein structures were used: (1) the homology model of the human P-gp (*h*P-gp) in an inward-facing conformation, constructed as previously reported [36]; (2) the recently published cryo-EM structure of encequidar bound to the *h*P-gp (PDB code 7O9W) [35] that represents an occluded conformation of the transporter. The docking score, extra precision glide score, and induced-fit docking score values were calculated and used to rank the obtained ligand-protein complexes. Additionally, the molecular mechanics/generalized Born surface area (MM/GBSA) approach was applied to estimate the free energy of binding of **11** and verapamil to individual conformations of *h*P-gp.

The binding sites of **11** and verapamil in *h*P-gp predicted by IFD were located in the upper part of the internal binding cavity (Figure 3A,B), similarly to other P-gp inhibitors seen in X-ray or cryo-EM complexes [33,34,35]. For verapamil, the binding modes found in docking to the homology model and to the cryo-EM structure were similar and followed the protein-ligand interaction pattern reported for verapamil by other authors (Figure 4A,B) [37,38,39]. The top-ranked docking poses of verapamil in both protein structures formed hydrogen bonds with Tyr307 and Tyr953 as well as aromatic π-π interactions with Phe983. The calculated free energy of binding (ΔG) of verapamil to *h*P-gp was −87.19 kcal/mol and −102.10 kcal/mol for the homology model and structure 7O9W, respectively (Appendix A).

The IFD results for **11** revealed that the predominant number of observed top-ranked poses of this compound docked to the *h*P-gp homology model represented one out of two variants—pose **Ia** or **Ib**, illustrated in Figure 5A. In both variants, the triphenylamine moiety of **11** was trapped inside the hydrophobic pocket built by residues of TM1, TM6, TM7, and TM12 and formed multiple π-π stacking contacts with some of the surrounding phenylalanines. The carboxylic group of **11** anchored the molecule by hydrogen bonding with a polar residue, mainly Gln946 (pose **Ia**) or Gln347 (pose **Ib**). Substantial binding interactions observed for both variants of pose **I** resulted in high docking scores and free binding energy: −87.96 kcal/mol and −68.07 kcal/mol for the best docking poses **Ia** and **Ib**, respectively (Appendix A).

During IFD docking of **11** to the cryo-EM structure 7O9W, a docking pose **II** was observed predominantly among the high-scored results (ΔG = −92.06 kcal/mol for the best docking score, Figure 5B). Interestingly, despite a different, occluded conformation of the protein in 7O9W, caused by kinked transmembrane helices TM4 and TM10, the binding mode observed for pose **II** resembled the interaction pattern of **Ib**—the triphenylamine moiety of **11** was located inside the phenylalanine region of TM1, TM6, TM7 and TM12, and involved in multiple π-π interactions, while the carboxylic end of the ligand formed the hydrogen bond with Gln347.

It should be noted that all of the discussed binding modes of **11** predicted by IFD were consistent, both in terms of the observed ligand-protein interactions and free energy of binding. Analysis of the interaction pattern for **11** showed a clear overlap with the binding mode seen in the present IFD results for verapamil and was highly in accordance with the experimental mutagenesis data on the verapamil binding site [40,41], suggesting a similar mechanism of inhibition of the rhodamine 123 efflux for both compounds.

Multiple in silico studies were performed for structurally diverse compounds that act as P-gp substrates and inhibitors. In general, P-gp inhibitors tend to be highly lipophilic molecules with a higher log P value and a higher molecular weight compared to P-gp substrates. The main impact on interactions between the inhibitor and the P-gp binding site usually comes from a large number of hydrophobic and van der Waals contacts formed by the aryl, alkyl, and other lipophilic groups of the ligand [8,10,12,33,42]. Functional groups such as carbonyl, ether, or a tertiary amine are also often present in the structure of P-gp inhibitors, allowing hydrogen bond formation or ionic interactions with the protein. However, to the best of our knowledge, the literature has not reported any cases of increasing the modulatory action of P-gp by introducing the carboxyl group to a molecule. In fact, the carboxylic acid group is not considered favorable for binding to the ABCB1 transporter and generally does not occur in the structures of known P-gp inhibitors. Our docking studies show that a carboxylic group, despite ionization at physiological pH, can successfully interact with residues of the P-gp binding site and thus can contribute to inhibitor binding.

### 2.5. Lipophilicity

In search of answers to the observed results of different influences of the carboxyl group on bioactivity, we have decided to estimate selected physicochemical properties, namely lipophilicity (log P) and solubility (log S). The obtained results of predicted measures (Appendix A) [43] showed comparable solubility of all investigated compounds, but their lipophilicity is more diverse. For two series, **1–4** and **5–8,** the introduction of the carboxyl group affects the values of log P more significantly (increase from 3.69 to 4.67 and from 3.78 to 4.15), while for the third series **9–12,** such an impact of the carboxyl group is not observed (values of log P in the range of 5.19 to 5.33). The lipophilicity of **9** is so high that the introduction of the carboxyalkyl chain does not play a significant role in changing it.

## 3. Materials and Methods

### 3.1. Chemistry

All reagents for syntheses were purchased from Merck KGaA (Darmstadt, Germany) and used without further purification. Melting point (uncorrected) has been determined on the Boetius apparatus (Carl Zeiss Jena). The IR spectrum has been recorded with Jasco FT IR-670 Plus spectrophotometer (JASCO Corp., Tokyo, Japan) in the KBr disk (1 mg sample/400 mg KBr). The MS analyses were obtained on the AmaZon ETD mass spectrometer (Bruker Daltonics, Bremen, Germany). Scan parameters: scan range 100–1000 *m*/*z*, positive ionization mode. CID fragmentation was in the ion trap analyzer with the aid of helium gas. The collision energy was set to ca. 1 eV. The samples were introduced into the mass spectrometer in a CH_3_OH:CHCl_3_ 1:1 solution with 0.1% HCOOH acidification. The ^1^H-NMR spectra and ^13^C-NMR spectra were recorded on a JEOL JNM-ECZR500 RS1 (ECZR version) at 500 and 126 MHz, respectively, and were reported in parts per million (ppm) using deuterated solvent for calibration (DMSO-*d*_6_). The coupling constants values (J) were reported in hertz (Hz).

#### 3.1.1. General Method for the Synthesis of Compounds **1–8** and **10–12** [24]

Two mmol of rhodanine or the appropriate rhodanine-3-carboxyalkyl acid, 3 g of 4-Å molecular sieves, 20 mL of isopropyl alcohol, 2 mmol of the appropriate aldehyde, and 1.4 mL (5 eq) of triethylamine were placed in the round bottom flask. The reaction mixture was heated under reflux for 5 h. After heating, the hot solution was filtered. The filtrate was cooled, and 2M hydrochloric acid was added thereto until the solution was acidic. The obtained precipitate was filtered off and crystallized from glacial acetic acid.

#### 3.1.2. Preparation of Compound **9** [44]

Five mmol of rhodanine, 7 g of 4Å molecular sieves, 30 mL of isopropyl alcohol, 5 mmol of the appropriate aldehyde, and 3.5 mL (5 eq) of triethylamine were placed in the round bottom flask. The reaction mixture was heated under reflux for 6 h. After heating, the hot solution was filtered. The filtrate was cooled, and 2 M hydrochloric acid was added thereto until the solution was acidic. The resulting mixture was extracted with two 25 mL portions of chloroform. The organic fractions were combined and washed with three 25 mL portions of water. The solution was dried over anhydrous magnesium sulfate, filtered, and chloroform was evaporated from the filtrate. The obtained precipitate was crystallized from isopropanol.

5-([1,1′-biphenyl]-4-ylmethylene)-2-thioxothiazolidin-4-one (**1**)

Yield 85.2%, m.p. 239–241 °C, MS [M+1]^+^ 298, IR (1 mg/400 mg KBr) cm^−1^: 1719.23 N-CO-C, 1705.73 C=O conj., 1642.09 C=C exo., 1173.47 C=S; ^1^H-NMR (500 MHz, DMSO-*d*_6_) δ 13.82 (s, 1H, H-N), 7.80 (d, *J* = 8.3 Hz, 2H, C3′H, C5′H), 7.70 (d, *J* = 7.2 Hz, 2H, C2″H, C6″H), 7.63 (d, *J* = 8.6 Hz, 3H, C=CH-, C2′H, C6′H), 7.45 (t, *J* = 7.6 Hz, 2H, C3″H, C5″H), 7.37 (t, *J* = 7.3 Hz, 1H, C4″H); ^13^C-NMR (126 MHz, DMSO-*d*_6_) δ 196.0 (C=S), 169.9 (N-C=O), 142.6 (C4′-Ar), 139.3 (Ar-C1″), 132.5 C=CH-Ar, 131.7 (C1′), 129.6 (C3′, C6″), 128.9 (C4″), 128.0 (C2′, C6′), 127.4 (C2″, C6″), 125.8 (C=CH-Ar).

2-(5-([1,1′-biphenyl]-4-ylmethylene)-4-oxo-2-thioxothiazolidin-3-yl)acetic acid (**2**)

Yield 64.0%, m.p. 257–258 °C, MS [M+1]^+^ 356, IR (1 mg/400 mg KBr) cm^−1^: 1735.62 N-CO-C, 1711.51 C=O conj., 1641.13 C=C exo., 1199.51 C=S; ^1^H-NMR (500 MHz, DMSO-*d*_6_) δ 13.42 (s, 1H, HOOC-), 7.89 (s, 1H, C=CH-Ar), 7.83 (d, *J* = 8.6 Hz, 2H, C3′H, C5′H), 7.73–7.70 (m, 4H, C2′H, C6′H, C2″H, C6″H), 7.46 (t, *J* = 7.6 Hz, 2H, C3″H, C5″H), 7.39 (t, *J* = 7.3 Hz, 1H, C4″H), 4.71 (s, 2H, HOOC-CH_2_-); ^13^C-NMR (126 MHz, DMSO-*d*_6_) δ 193.6 (C=S), 167.9 (HOOC-), 166.9 (N-C=O), 143.0 (C4′), 139.2 (Ar-C1″), 134.1 (C3′, C5′), 132.4 (C1′), 132.1 (C=CH-Ar), 129.7 (C3″, C5″), 129.0 (C4″), 128.1 (C2′, C6′), 127.4 (C2″, C6″), 122.0 (C=CH-Ar), 45.6 (HOOC-CH_2_-).

2-(5-([1,1′-biphenyl]-4-ylmethylene)-4-oxo-2-thioxothiazolidin-3-yl)propionic acid (**3**)

Yield 97.2%, m.p. 253–255 °C, MS [M+1]^+^ 370, IR (1 mg/400 mg KBr) cm^−1^: 1737.55 N-CO-C, 1707.66 C=O conj., 1641.13 C=C exo., 1171.54 C=S; ^1^H-NMR (500 MHz, DMSO-*d*_6_) δ 12.51 (s, 1H, HOOC-), 7.81–7.79 (m, 3H, C=CH-, C3′H, C5′H), 7.70 (d, *J* = 7.4 Hz, 2H, C2″H, C6″H), 7.66 (d, *J* = 8.3 Hz, 2H, C2′H, C6′H), 7.45 (t, *J* = 7.6 Hz, 2H, C3″H, C5″H), 7.38 (t, *J* = 7.3 Hz, 1H, C4″H), 4.19 (t, *J* = 7.7 Hz, 2H, HOOC-CH_2_-CH_2_-), 2.60 (t, *J* = 7.9 Hz, 2H, HOOC-CH_2_-CH_2_-); ^13^C-NMR (126 MHz, DMSO-*d*_6_) δ 193.7 (C=S), 172.3 (HOOC-), 167.2 (N-C=O), 142.8 (C4′-Ar), 139.2 (Ar-C1″), 133.0 (C3′, C5′), 132.5 (C=CH-Ar), 131.9 (C1′), 129.7 (C3″, C5″), 128.9 (C4″), 128.1 (C2′, C6′), 127.4 (C2″, C6″), 122.7 (C=CH-Ar), 40.5 (HOOC-CH_2_-CH_2_-), 31.3 (HOOC-CH_2_-CH_2_-). 

2-(5-([1,1′-biphenyl]-4-ylmethylene)-4-oxo-2-thioxothiazolidin-3-yl)butyric acid (**4**)

Yield 57.3%, m.p. 197–198 °C, MS [M+1]^+^ 384, IR (1 mg/400 mg KBr) cm^−1^: 1736.58 N-CO-C, 1712.48 C=O conj., 1641.13 C=C exo., 1167.69 C=S; ^1^H-NMR (500 MHz, DMSO-*d*_6_) δ 12.11 (s, 1H, HOOC-), 7.82 (d, *J* = 8.3 Hz, 2H, C3′H, C5′H), 7.78 (s, 1H, C=CH-Ar), 7.71 (d, *J* = 7.2 Hz, 2H, C2″H, C6″H), 7.67 (d, *J* = 8.3 Hz, 2H, C2′H, C6′H), 7.46 (t, *J* = 7.6 Hz, 2H, C3″H, C5″H), 7.38 (t, *J* = 7.3 Hz, 1H, C4″H), 4.04 (t, *J* = 6.9 Hz, 2H, HOOC-CH_2_-CH_2_-CH_2_), 2.27 (t, *J* = 7.2 Hz, 2H, HOOC-CH_2_-CH_2_-CH_2_-), 1.89–1.83 (m, 2H, HOOC-CH_2_-CH_2_-CH_2_-); ^13^C-NMR (126 MHz, DMSO-*d*_6_) δ 194.0 (C=S), 174.3 (HOOC-), 167.7 (N-C=O), 142.7 (C4′-Ar), 139.2 (Ar-C1″), 132.8 (C=CH-Ar), 132.6 (C1′), 131.9 (C3′, C5′), 129.7 (C3″, C5″), 128.9 (C4″), 128.1 (C2′, C6′), 127.4 (C2″, C6″), 122.8 (C=CH-Ar), 44.3 (HOOC-CH_2_-CH_2_-CH_2_-), 31.5 (HOOC-CH_2_-CH_2_-CH_2_-), 22.5 (HOOC-CH_2_-CH_2_-CH_2_-).

5-(4-(benzyloxy)benzylidene)-2-thioxothiazolidin-4-one (**5**)

Yield 74.8%, m.p. 232–234 °C, MS [M-1]ˉ 326, IR (1 mg/400 mg KBr) cm^−1^: 1723.09 N-CO-C, 1711.51 C=O conj., 1643.05 C=C exo., 1258.32 O-CH_2_-, 1172.51 C=S; ^1^H-NMR (500 MHz, DMSO-*d*_6_) δ 13.72 (s, 1H), 7.56 (s, 1H, C=CH-Ar), 7.52 (d, *J* = 8.6 Hz, 2H, C2H, C6H), 7.42 (d, *J* = 7.7 Hz, 2H, C2′H, C6′H), 7.36 (t, *J* = 7.6 Hz, 2H, C3′H, C5′H), 7.30 (t, *J* = 7.2 Hz, 1H, C4′H), 7.14 (d, *J* = 8.6 Hz, 2H, C3H, C5H), 5.17 (d, *J* = 16.6 Hz, 2H, O-CH_2_-Ar); ^13^C-NMR (126 MHz, DMSO-*d*_6_) δ 196.1 (S=C-S), 170.1 (N-C=O), 160.9 (C4 Ar), 137.0 (C1′ Ar), 133.2 (C2, C6 Ar), 132.3 (=CH-Ar), 129.0 (C3′, C5′ Ar), 128.6 (C2′, C6′ Ar), 128.4 (C4′ Ar), 126.2 (C=CH-Ar), 123.0 (C1 Ar), 116.4 (C3, C5 Ar), 70.1 (O-CH_2_-Ar).

2-(5-(4-(benzyloxy)benzylidene)-4-oxo-2-thioxothiazolidin-3-yl)acetic acid (**6**) [45]

Yield 61.5%, m.p. 211–213 °C, MS [M+1]^+^ 386, IR (1 mg/400 mg KBr) cm^−1^: 1731.76 N-CO-C, 1710.55 C=O conj., 1638.23 C=C exo., 1265.07 O-CH_2_-, 1176.36 C=S; ^1^H-NMR (500 MHz, DMSO-*d*_6_) δ 13.42 (s, 1H, HOOC-), 7.80 (s, 1H, C=CH-Ar), 7.60 (d, *J* = 8.9 Hz, 2H, C2H, C6H), 7.44–7.42 (m, 2H, C2′H, C6′H), 7.38–7.35 (m, 2H, C3′H, C5′H), 7.32–7.29 (m, 1H, C4′H), 7.17–7.15 (m, 2H, C3H, C5H), 5.16 (s, 2H, O-CH_2_-Ar), 4.69 (s, 2H, HOOC-CH_2_); ^13^C-NMR (126 MHz, DMSO-*d*_6_) δ 193.6 (S=C-S), 167.9 (HOOC-), 167.0 (N-C=O), 161.4 (C4 Ar), 136.9 (C1′ Ar), 134.6 (=CH-Ar), 133.6 (C2, C6 Ar), 129.1 (C3′, C5′ Ar), 128.6 (C4′ Ar), 128.4 (C2′, C6′ Ar), 126.1 (C=CH-Ar), 119.1 (C1 Ar), 116.5 (C3, C5 Ar), 70.1 (O-CH_2_-Ar), 45.5 (HOOC-CH_2_).

2-(5-(4-(benzyloxy)benzylidene)-4-oxo-2-thioxothiazolidin-3-yl)propionic acid (**7**)

Yield 85.9%, m.p. 190–192 °C, MS [M+1]^+^ 400, IR (1 mg/400 mg KBr) cm^−1^: 1738.51 N-CO-C, 1711.51 C=O conj., 1642.09 C=C exo., 1260.25 O-CH_2_-, 1171.54 C=S; ^1^H-NMR (500 MHz, DMSO-*d*_6_) δ 12.47 (s, 1H, HOOC-), 7.72 (s, 1H, C=CH-Ar), 7.56 (d, *J* = 8.9 Hz, 2H, C2H, C6H), 7.44–7.42 (m, 2H, C2′H, C6′H), 7.38–7.35 (m, 2H, C3′H, C5′H), 7.32–7.29 (m, 1H, C4′H), 7.16–7.14 (m, 2H, C3H, C5H), 5.16 (s, 2H, O-CH_2_-Ar), 4.18 (t, *J* = 7.9 Hz, 2H, HOOC-CH_2_CH_2_), 2.61–2.57 (m, 2H, HOOC-CH_2_CH_2_); ^13^C-NMR (126 MHz, DMSO-*d*_6_) δ 193.6 (S=C-S), 172.3 (HOOC-), 167.3 (N-C=O), 161.2 (C4 Ar), 136.9 (C1′ Ar), 133.7 (C2, C6 Ar), 133.5 (=CH-Ar), 129.0 (C3′, C5′ Ar), 128.6 (C4′ Ar), 128.4 (C2′, C6′ Ar), 126.2 (C=CH-Ar), 119.7 (C1 Ar), 116.5 (C3, C5 Ar), 70.1 (O-CH_2_-Ar), 40.5 (HOOC-CH_2_CH_2_), 31.3 (HOOC-CH_2_CH_2_).

2-(5-(4-(benzyloxy)benzylidene)-4-oxo-2-thioxothiazolidin-3-yl)butyric acid (**8**)

Yield 42.3%, m.p. 165–167 °C, MS [M+1]^+^ 414, IR (1 mg/400 mg KBr) cm^−1^: 1738.50 N-CO-C, 1710.01 C=O conj., 1643.31 C=C exo., 1262.18 O-CH_2_-, 1173.44 C=S; ^1^H-NMR (500 MHz, DMSO-*d*_6_) δ 12.10 (s, 1H, HOOC-), 7.71 (s, 1H, C=CH-Ar), 7.56 (d, *J* = 8.9 Hz, 2H, C2H, C6H), 7.43 (d, *J* = 7.2 Hz, 2H, C2′H, C6′H), 7.38–7.35 (m, 2H, C3′H, C5′H), 7.32–7.29 (m, 1H, C4′H), 7.15 (d, *J* = 8.9 Hz, 2H, C3H, C5H), 5.16 (s, 2H, O-CH_2_-Ar), 4.03 (t, *J* = 7.0 Hz, 2H, HOOC-CH_2_CH_2_CH_2_), 2.25 (t, *J* = 7.2 Hz, 2H, HOOC-CH_2_CH_2_CH_2_), 1.87–1.82 (m, 2H, HOOC-CH_2_CH_2_CH_2_); ^13^C-NMR (126 MHz, DMSO-*d*_6_) δ 194.0, 174.2 (HOOC-), 167.7 (N-C=O), 161.1 (C4 Ar), 136.9 (C1′ Ar), 133.5 (C2, C6 Ar), 133.4 (=CH-Ar), 129.0 (C3′, C5′ Ar), 128.6 (C4′ Ar), 128.4 (C2′, C6′ Ar), 126.3 (C=CH-Ar), 119.8 (C1 Ar), 116.4 (C3, C5 Ar), 70.1 (O-CH_2_-Ar), 44.2 (HOOC-CH_2_CH_2_CH_2_), 31.5 (HOOC-CH_2_CH_2_CH_2_), 22.5 (HOOC-CH_2_CH_2_CH_2_).

5-(4-(diphenylamino)benzylidene)-2-thioxothiazolidin-4-one (**9**)

Yield 25.0%, m.p. 237–239 °C, MS [M+1]^+^ 375, IR (1 mg/400 mg KBr) cm^−1^: 1720.19 N-CO-C, 1682.59 C=O conj., 1643.05 C=C exo., 1174.44 C=S; ^1^H-NMR (500 MHz, DMSO-*d*_6_) δ 13.65 (s, 1H, H-N), 7.46 (s, 1H, C=CH-Ar), 7.36 (d, *J* = 8.9 Hz, 2H, C3′H, C5′H), 7.32 (t, *J* = 7.9 Hz, 4H, C3″H, C5″H, C3″′H, C5″′H), 7.13 (t, *J* = 7.4 Hz, 2H, C4″H, C4″′H), 7.07 (d, *J* = 7.7 Hz, 4H, C2″H, C6″H, C2″′H, C6″′H), 6.85 (d, *J* = 8.9 Hz, 2H, C2′H, C6′H); ^13^C-NMR (126 MHz, DMSO-*d*_6_) δ 195.8, 170.0, 150.2, 146.2, 132.9, 132.3, 130.4, 126.4, 125.6, 125.4, 121.8, 120.3.

### 3.2. Biological Assays

#### 3.2.1. Cell Lines

L5178Y mouse T-cell lymphoma cells (PAR) (ECACC Cat. No. 87111908, obtained from FDA, Silver Spring, MD, USA) were transfected with pHa MDR1/A retrovirus, as previously described by Cornwell et al. [46]. The ABCB1-expressing cell line (MDR) was selected by culturing the infected cells with colchicine. The cell lines were cultured in McCoy′s 5A medium (Sigma-Aldrich, St. Louis, MO, USA) as described previously [47].

#### 3.2.2. Modulation of P-gp: Rhodamine 123 Accumulation Assay

The inhibition of the MDR efflux pump P-gp (ABCB1) was detected by the retention of rhodamine 123 by ABCB1 (P-glycoprotein) in MDR mouse T-lymphoma cells using flow cytometry according to the protocol published by the former study [47]. The fluorescence of the cell population was evaluated using a Partec CyFlow^®^ flow cytometer (Partec, Münster, Germany), measuring 10,000 individual cells in the given sample. 

#### 3.2.3. Cytotoxicity and Antiproliferative Assays

The cytotoxic and antiproliferative effect of the derivatives on parental (PAR) and multidrug-resistant (MDR) mouse T-lymphoma cells was assessed by MTT assay [36,48]. The compounds were diluted by 2-fold serial dilution in a volume of 100 μL medium. Then, in the case of the cytotoxicity assay, 10^4^ cells in 100 μL of medium and in the case of the antiproliferative assay, 6 × 10^3^ cells in 100 μL medium were pipetted to each well, with the exception of the medium control wells. The plates were incubated at 37 °C for 24 h or 72 h, respectively; at the end of the incubation period, 20 μL of MTT (thiazolyl blue tetrazolium bromide, Sigma) solution (from a 5 mg/mL stock) were added to each well as described previously [36,48]. 

The results are expressed in terms of IC**_50_**, defined as the inhibitory dose that reduces by 50% the growth of the cells exposed to the tested compound. The results are based on two independent assays with four parallel samples in each assay.

#### 3.2.4. Drug Combination Assay

Doxorubicin (2 mg/mL, Teva Pharmaceuticals, Budapest, Hungary) was serially diluted in the horizontal direction at 100 µL as previously described [48], starting with 17.2 µM. The derivatives were subsequently diluted in the vertical direction, and the starting concentration was determined based on the IC_50_ of the antiproliferative assay. The dilutions of the derivatives were prepared vertically in the microtiter plate in a 50 µL volume. The cells were resuspended in culture medium and distributed into each well in 50 µL containing 6 × 10^3^ cells, with the exception of the medium control wells, to a final volume of 200 µL per well. The plates were incubated for 72 h at 37 °C in a CO_2_ incubator, and at the end of the incubation period, the cell growth was determined by the MTT staining method, as described earlier. Drug interactions were evaluated using CompuSyn software [49]. The dose-response curve of the compounds and their combinations was fit to a linear model using the median effect equation in order to obtain the median effect value (corresponding to the IC_50_) and slope (m). The type of interaction between drugs was calculated using the combination index, in which a CI value close to 1 shows additivity, while a CI < 1 means synergism, and a CI > 1 refers to an antagonism.

### 3.3. Crystallographic Study

Single crystals were obtained by slow evaporation of the solvent under ambient conditions from propan-2-ol for **3**, a mixture of ethanol and 1,2-dichlorobenzene (1:1, *v/v*) for **7,** and ethanol for **11**. Intensity data were collected on the Oxford Diffraction SuperNova four circle diffractometer for **3**, while for **7** and **11** on the Bruker-Nonius Kappa CCD four circle diffractometer, equipped with the Mo (0.71069 Å) Kα radiation source and graphite monochromator. The phase problems were solved by direct methods using SIR-2014 [50] for **3** and SHELXS [51] for **7** and **11** programs. The parameters of the obtained models were refined by full-matrix least-squares on F^2^ using SHELXL [51] program. All non-hydrogen atoms were refined anisotropically. The hydrogen atoms attached to carbon atoms were refined isotropically in calculated positions using the riding model with U_iso_(H) fixed at 1.2 U_eq_ of C, while the hydrogen atoms attached to oxygen atoms were located in the difference Fourier map and refined without any restraints. For molecular graphics, the MERCURY program was used [52].

#### Crystallographic Data

**3**: C_19_H_15_NO_3_S_2_, M_r_ = 369.44, triclinic, space group P-1, a = 9.7451(4) Å, b = 13.9145(5) Å, c = 14.3358(5) Å, α = 101.765(3)°, β = 90.557(1)°, γ = 101.125(3)°, V = 1717.1(1) Å^3^, Z = 4, T = 120(1)K, 24118 reflections collected, 8022 unique reflections (R_int_ = 0.0349), R1 = 0.0426, wR2 = 0.0942 [I > 2σ(I)], R1 = 0.0650, wR2 = 0.1074 (all data).

**7**: C_20_H_17_NO_4_S_2_, M_r_ = 399.46, triclinic, space group P-1, a = 11.1930(3) Å, b = 13.2410(4) Å, c = 13.9940(4) Å, α = 113.250(2)°, β = 93.208(2)°, γ = 101.598(2)°, V = 1845.5(1) Å^3^, Z = 4, T = 293(2)K, 14691 reflections collected, 8425 unique reflections (R_int_ = 0.0254), R1 = 0.0481, wR2 = 0.1149 [I > 2σ(I)], R1 = 0.0626, wR2 = 0.1223 (all data).

**11**: C_22_H_20_N_2_O_3_S_2_, M_r_ = 460.55, triclinic, space group P-1, a = 7.9420(2) Å, b = 9.5630(3) Å, c = 16.0670(4) Å, α = 77.790(2)°, β = 76.769(2)°, γ = 69.832(1)°, V = 1103.25(5) Å^3^, Z = 2, T = 100(2)K, 8969 reflections collected, 5028 unique reflections (R_int_ = 0.0279), R1 = 0.0358, wR2 = 0.0816 [I > 2σ(I)], R1 = 0.0508, wR2 = 0.0889 (all data).

CCDC 2174182-2174184 contains the supplementary crystallographic data. These data can be obtained free of charge from The Cambridge Crystallographic Data Centre via www.ccdc.cam.ac.uk/data_request/cif (accessed on 22 May 2022).

### 3.4. Molecular Docking

The cryo-EM structure of the human P-gp bound to encequidar was retrieved from the PDB database (PDB code 7O9W). The homology model of the human P-gp was built by multiple comparative modeling using X-ray structures 4Q9H, 4XWK, and 4M1M of the murine P-gp as templates) [36]. The sequence of the human P-gp multidrug transporter (accession no. P08183) was retrieved from UniProtKB/TrEMBL database [53]. 

Flexible receptor docking of compound **11** and verapamil was performed in Schrödinger Suite [27]. Both protein structures—the homology model as well as the 7O9W structure—before docking were subjected to the Protein Preparation Wizard procedure (minimization of energy using OPLS2005 force field). The structure of **11**, obtained in this work under crystallographic studies, was used as the initial conformation of the ligand. The protonation states of the **11** and verapamil were generated in the LigPrep module to give neutral or ionized structures that were further optimized by minimization of energy, using options of the MacroModel module (OPLS2005 force field, no solvent) 

An induced-fit docking study was performed according to the protocol implemented in Schrödinger Suite. In the case of compound **11**, the receptor grid box was defined as a large binding cavity limited by the transmembrane region of the *h*P-gp, reported for the P08183 sequence. For verapamil, the putative binding site was built around a centroid of P-gp residues found by Loo in the mutagenesis experiments to interact with this drug [40,41]. During the procedure, the conformation of the residue side chains at the distance of 6Å from the ligand was predicted and optimized. The resulting ligand-protein complexes were used as temporary templates for the re-docking of the ligands with higher Glide XP precision. The obtained results were ranked on the basis of the docking score functions: extra precision glide score (XP Gscore) and induced-fit docking score (IFD score). Additionally, the molecular mechanics/generalized Born surface area (MM/GBSA) method was used to estimate the free energy of the binding of **11** and verapamil to individual conformations of *h*P-gp.

The graphic presentation of selected protein-ligand complexes with the highest docking scores was prepared in the PyMOL software, v. 2.5.2 [54].

## 4. Conclusions

The present work demonstrated the previously unknown biological activity of rhodanine derivatives and the influence of the carboxyl group in their structure on this activity. Importantly, **10–12** (substituted with the carboxyl group) exhibit greater potential than **9** (no carboxyl substituent) to inhibit the P-gp. The obtained results indicate derivatives of rhodanine as a new group of compounds for medicinal chemistry and drug design in the search for new inhibitors of efflux pump P-gp.

An emerging research strategy could be the application of efflux pump inhibitors (EPIs) and their use as adjuvant compounds or chemosensitizers to improve the outcome of antitumor therapy by co-administering them with chemotherapeutic agents. For this reason, the ability of the compounds to inhibit ABCB1 should be investigated; furthermore, their interaction with standard chemotherapeutic drugs should be evaluated. Based on our results, potent ABCB1 modulators have been found; however, their combination with doxorubicin was not synergistic. They cannot be applied in combination with doxorubicin except for compound **8**, showing additive interaction. 

## Data Availability

All data generated or analyzed during these studies are included in this published article. Moreover, the datasets used and/or analyzed during the current study are available from the corresponding authors upon reasonable request.

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
