# Peer review of "5-Arylidenerhodanines as P-gp Modulators: An Interesting Effect of the Carboxyl Group on ABCB1 Function in Multidrug-Resistant Cancer Cells"

_ijms, 2022, doi:10.3390/ijms231810812_

Round 1

Reviewer 1 Report

The manuscript describes the effect of newly synthesized rhodanine-derivates on mouse T-lymphoma cells (parental and overexpressing P-glycoprotein) and the docking of one of the compounds in structural models of P-glycoprotein. My main concern about the manuscript is the lack of statistical information: there is not mention of the number of independent experiments performed or any statistical analysis of the data. Where the experiments in cells done more than one time? How many times? Is the SD between replicates of one experiment or between the values of multiple independent experiments? If the authors don't mention the number of experiments anywhere in the manuscript, we can asume that the experiments were done only once, which is not scientifically correct. If the authors indeed performed multiple experiments and omitted the information in the text, they should make it clear so the results can be scientifically acceptable. 

The criteria for the analysis of the data is not clear/consistent. Lines 112 to 114, “Most of the compounds under these conditions did not show significant cytotoxic effects against two tested cell lines, except for 3 (IC50 = 13.88 μM and 17.02 μM for PAR and MDR cells, respectively)” . However, in lines 125 to 127 says: “To evaluate the ability of target rhodanines to resensitize MDR cells to the anticancer drug doxorubicin, interactions between doxorubicin and compounds showing efficient cytotoxic activity (1, 3, 5 and 8) were evaluated using a checkerboard combination assay”. Also, can the authors show data confirming that the MDR cells are indeed overexpressing P-gp?

How do the authors define the “most active efflux modulator” mentioned in Line 169?  The authors mentioned in the introduction that they are evaluating the inhibitory effect of the compounds on ABCB1-efflux (line 60), but there is not data or discussion about efflux in the manuscript.

In general, the discussion of the data needs to be improved.

Line 22, it would be useful to mention that you synthesized the compounds.

Line 23, add mouse: mouse T-lymphoma cancer cells and remove “mouse” from line 26.

Line 28, add “tested” : 11, one of the most potent P-gp inhibitors tested here or in this study.

Line 37, what reference was used to say that “More than 40 ABC transporters have been identified in humans …”? That number does not seem to agree with published information.

Lines 39 and 43,  remove “the” from “the P-glycoprotein”

Remove “it” from “The filtrate was cooled and 2M hydrochloric acid was added thereto until the solution it was acidic” in lines 287 and 294.

Please, include proper citations to methods. Some citations are missing.

Author Response

We thank Reviewers for the constructive comments. Enclosed, please, find our answers:

Review 1

The manuscript describes the effect of newly synthesized rhodanine-derivates on mouse T-lymphoma cells (parental and overexpressing P-glycoprotein) and the docking of one of the compounds in structural models of P-glycoprotein. My main concern about the manuscript is the lack of statistical information: there is not mention of the number of independent experiments performed or any statistical analysis of the data. Where the experiments in cells done more than one time? How many times? Is the SD between replicates of one experiment or between the values of multiple independent experiments? If the authors don't mention the number of experiments anywhere in the manuscript, we can asume that the experiments were done only once, which is not scientifically correct. If the authors indeed performed multiple experiments and omitted the information in the text, they should make it clear so the results can be scientifically acceptable. 

It has been added to the Materials and Methods:

„The results are based of two independent assays with four parallel samples in each assay.”

The criteria for the analysis of the data is not clear/consistent. Lines 112 to 114, “Most of the compounds under these conditions did not show significant cytotoxic effects against two tested cell lines, except for 3 (IC50 = 13.88 μM and 17.02 μM for PAR and MDR cells, respectively)” . However, in lines 125 to 127 says: “To evaluate the ability of target rhodanines to resensitize MDR cells to the anticancer drug doxorubicin, interactions between doxorubicin and compounds showing efficient cytotoxic activity (1, 3, 5 and 8) were evaluated using a checkerboard combination assay”. Also, can the authors show data confirming that the MDR cells are indeed overexpressing P-gp?

We thank Reviewer for the comment, it was a mistake in the paragraph. The sentence in lines 125-127 has been corrected:

“To evaluate the ability of target rhodanines to resensitize MDR cells to the anticancer drug doxorubicin, interactions between doxorubicin and compounds showing efficient cytotoxic and antiproliferative activities (1, 3, 5 and 8) were evaluated using a checkerboard combination assay.”

The MDR cell line is an artificial cell line overexpressing the P-gp as described previously:

Galski H, Lazarovici P, Gottesman MM, Murakata C, Matsuda Y, Hochman J. KT-      5720 reverses multidrug resistance in variant S49 mouse lymphoma cells transduced      with the human MDR1 cDNA and in human multidrug-resistant carcinoma cells. Eur. J. Cancer. 1995;31A(3):380-8. doi:10.1016/0959-8049(94)00511-3. PMID:7786606.

The same cell line was applied together with its sensitive parental cell line in several studies:

 - Spengler G, Viveiros M, Martins M, Rodrigues L, Martins A, Molnar J, Couto I, Amaral L. Demonstration of the activity of P-glycoprotein by a semi-automated fluorometric method. Anticancer Res. 2009 Jun;29(6):2173-7. PMID: 19528478.

  - Handzlik J, Spengler G, Mastek B, Dela A, Molnar J, Amaral L, Kieć-Kononowicz.5-arylidene(thio)hydantoin derivatives as modulators of cancer efflux pump. Acta Pol Pharm. 2012 Jan-Feb;69(1):149-56. PMID: 22574520.

- Bourichi S, Misbahi H, Rodi YK, Chahdi FO, Essassi EM, Szabó S, Szalontai B, Gajdács M, Molnár J, Spengler G. In Vitro Evaluation of the Multidrug Resistance Reversing Activity of Novel Imidazo[4,5-b]pyridine Derivatives. Anticancer Res. 2018 Jul;38(7):3999-4003. doi: 10.21873/anticanres.12687.

How do the authors define the “most active efflux modulator” mentioned in Line 169?  The authors mentioned in the introduction that they are evaluating the inhibitory effect of the compounds on ABCB1-efflux (line 60), but there is not data or discussion about efflux in the manuscript.

The “most active efflux modulator” has been changed to the “most active efflux pump modulator”. This term is more accurate because in the experiments the intracellular rhodamine 123 accumulation was measured, not the efflux of the fluorochrome.

In general, the discussion of the data needs to be improved.

To the “Conclusions” has been added the fragment:

An emerging research strategy could be the application of efflux pump inhibitors (EPIs) and their use as adjuvant compounds or chemosensitizers to improve the outcome of antitumor therapy, by co-administering them with chemotherapeutic agents. For this reason, the ability of the compounds to inhibit ABCB1 should be investigated, furthermore their interaction with standard chemotherapeutic drugs should be evaluated. Based on our results, potent ABCB1 modulators have been found, however their combination with doxorubicin was not synergistic, they cannot applied in combination with doxorubicin except for compound 8 showing additive interaction.

Line 22, it would be useful to mention that you synthesized the compounds.

We have added to the text “newly synthesized”

Line 23, add mouse: mouse T-lymphoma cancer cells and remove “mouse” from line 26.

The sentences have been corrected according to Reviewer’s suggestion.

Line 28, add “tested” : 11, one of the most potent P-gp inhibitors tested here or in this study.

The sentence has been corrected according to suggestion.

Line 37, what reference was used to say that “More than 40 ABC transporters have been identified in humans …”? That number does not seem to agree with published information.

The used reference “Genome Res.” 2001, 11, 1156-1166 was added to this section.

Lines 39 and 43,  remove “the” from “the P-glycoprotein”

It has been removed “the” in both sentences.

Remove “it” from “The filtrate was cooled and 2M hydrochloric acid was added thereto until the solution it was acidic” in lines 287 and 294.

It has been removed “it” in both sentences.

Please, include proper citations to methods. Some citations are missing.

It has been added to the manuscript the missing citations.

Reviewer 2 Report

Dear editor,

1.     This manuscript is interesting and well-done.

2.     The strength of this article is well organized for readers to understand for issue of MDR in refactory cancer.

3.     Inhibition of the ABCB1 efflux pump by Series of 5-arylidenerhodanines look so dramatic.

4.     But the regretting point is wishing it was more attentive about showing cell viabillity assay (e.g. dose dependant manner of 12). This article insist on ‘cytotoxicity’ or ‘ani-cancer effect’ by the 12, more detail figure about ‘cytotoxicity’ or ‘ani-cancer effect’ will have to be backed up. IC50 is just not enough. Conventionally, assy of the cytotoxic and anti-proliferate shows below figure.

Figure 1. Volume 2014 |Article ID 819548 | https://doi.org/10.1155/2014/819548

5.     Verapamil is known as calcium-channel blockers too. What do you think about the possibility of inhibition to calcium-channel blockers by12?

6.     It's just my opinion, this manuscript was accepted to be enough after english language and style are minor spell check.

Author Response

We thank Reviewers for the constructive comments. Enclosed, please, find our answers:

Review 2

  1. This manuscript is interesting and well-done.
  2. The strength of this article is well organized for readers to understand for issue of MDR in refactory cancer.
  3. Inhibition of the ABCB1 efflux pump by Series of 5-arylidenerhodanines look so dramatic.
  4. But the regretting point is wishing it was more attentive about showing cell viabillity assay (e.g. dose dependant manner of 12). This article insist on ‘cytotoxicity’ or ‘ani-cancer effect’ by the 12, more detail figure about ‘cytotoxicity’ or ‘ani-cancer effect’ will have to be backed up. IC50 is just not enough. Conventionally, assy of the cytotoxic and anti-proliferate shows below figure. Figure 1. Volume 2014 |Article ID 819548 | https://doi.org/10.1155/2014/819548

The IC50 values were evaluated using GraphPad software and the non-linear regression dose-dependence curves were calculated.

We have no possibility to create suggested figures. The necessary detail data was lost due to computer crash. The main focus of this manuscript is the P-gp inhibition and docking study.

  1. Verapamil is known as calcium-channel blockers too. What do you think about the possibility of inhibition to calcium-channel blockers by 12?

We have no possibility to investigate this compounds towards of calcium channel blockers.

  1. It's just my opinion, this manuscript was accepted to be enough after english language and style are minor spell check.

The whole text has been checked for English and corrected accordingly.

Round 2

Reviewer 2 Report

Thank you for your response and your kind words. 

My comments were indicted below. 

1. This manuscript is interesting and well-done. 

2. The strength of this article is well organized for readers to understand for issue of MDR in refractory cancer. 

3. The authors have addressed all my concerns. 

4. The revised manuscript has been significantly improved: necessary changes have been made to the text and figures, and the quality of "Materials and Methods" has been improved.